# Evaluation of Whole Brain Intravoxel Incoherent Motion (IVIM) Imaging

**DOI:** 10.3390/diagnostics14060653

**Published:** 2024-03-20

**Authors:** Kamil Lipiński, Piotr Bogorodzki

**Affiliations:** Institute of Radioelectronics and Multimedia Technology, Faculty of Electronics and Information Technology, Warsaw University of Technology, Nowowiejska 15/19, 00-665 Warsaw, Poland; piotr.bogorodzki@pw.edu.pl

**Keywords:** MRI, DWI, IVIM, *SNR*, perfusion, brain

## Abstract

Intravoxel Incoherent Motion (IVIM) imaging provides non-invasive perfusion measurements, eliminating the need for contrast agents. This work explores the feasibility of IVIM imaging in whole brain perfusion studies, where an isotropic 1 mm voxel is widely accepted as a standard. This study follows the validity of a time-limited, precise, segmentation-ready whole-brain IVIM protocol suitable for clinical reality. To assess the influence of *SNR* on the estimation of *S_0_*, *f*, *D**, and *D* IVIM parameters, a series of measurements and simulations were performed in MATLAB for the following three estimation techniques: segmented grid search, segmented curve fitting, and one-step curve fitting, utilizing known “ground truth” and noised data. Scanner-specific *SNR* was estimated based on a healthy subject IVIM MRI study in a 3T scanner. Measurements were conducted for 25.6 × 25.6 × 14.4 cm FOV with a 256 × 256 in-plane resolution and 72 slices, resulting in 1 × 1 × 2 mm voxel size. Simulations were performed for 36 *SNR* levels around the measured *SNR* value. For a single voxel grid, the search algorithm mean relative error *Ŝ*_0_, $\hat{f}$, ${\hat{D}}^{*}$, and $\hat{D}$ of at the expected *SNR* level were 5.00%, 81.91%, 76.31%, and 18.34%, respectively. Analysis has shown that high-resolution IVIM imaging is possible, although there is significant variation in both accuracy and precision, depending on *SNR* and the chosen estimation method.

## 1. Introduction

In many brain perfusion studies, contrast agents are used [1]. Although current gadolinium-based contrast agents have relatively low toxicity, the possibility of long-term, negative side effects is present [2]. A contrast-free MRI perfusion imaging method would improve clinical scanning protocols’ safety. Such a possibility arises when considering IVIM-based perfusion studies.

IVIM MRI was first described by Le Bihan in 1986 [3] as an extension of diffusion MRI (dMRI) that explores the random movement of molecules [4], particularly water, within biological tissues. This movement, known as diffusion, results from molecular collisions and follows a random walk pattern or so-called Brownian motion [5]. A dMRI sequence was first used, modified by Stejskal and Tanner [6] into a spin-echo sequence, where the MR echo signal was sensitized to water molecules’ diffusion speed by applying magnetic field gradient pulses. The resulting exponential decrease in the measured MR signal depends on the product of the diffusion coefficient (D) and the magnitude of sensitization, the so-called b-value (later in this work referred to as b or b-value).

In IVIM MRI, the movement of water, caused by the blood flow in the vessel network being aligned in multiple directions within a voxel, can be considered a fast (pseudo-diffusion, $D^{*}$) component. This pseudo-diffusion component and the extravascular component (regular diffusion, D) contribute to MR signal attenuation, allowing both tissue diffusion and blood microcirculation to be detected and separated [7]. Thus, the overall dMRI signal, as a combination of both fast and regular components, can be described using a bi-exponential signal decay model, given in Equation (1):(1)$$S\left(b\right)=S_{0}\cdot \left(f\cdot e^{-D^{*}\cdot b}+\left(1-f\right)\cdot e^{-D\cdot b}\right)$$
where:

S(b) represents the measured signal intensity in the DWI image for a given b.b stands for *b*-value—diffusion weighting factor, determined by the strength and timing of the diffusion gradients prior to signal echo.$S_{0}$ is the signal intensity in the absence of diffusion weighting (b=0).f represents the fraction of signal coming from quick-diffusing water molecules, which are assumed to be circulating in the blood, also referred to as the perfusion fraction.$D^{*}$ is the pseudo-diffusion coefficient, which reflects the diffusion of water in capillaries and small vessels.D represents the true diffusion coefficient, which characterizes the diffusion of extravascular water molecules.

An illustration of an example of the Intravoxel Incoherent Motion (IVIM) dataset is presented in Figure 1. The parameters chosen for this representation are $f=0.1,\;D=0.001,\;D^{*}=0.012,\;S_{0}=1$. These values were selected to demonstrate a clear and explanatory IVIM dataset generated in accordance with Equation (1).

While previous studies have extensively explored the applications of the IVIM technique in various medical fields, especially in pathological conditions [9,10,11], the influence of *SNR* on the accuracy of IVIM parameter determination remains insufficiently investigated in terms of clinical, time-limited whole brain scanning, only partially covered by simulations in silico [12]. In contrast to other studies conducted for IVIM-specific exams, which took most of the scanner time, this study aims to assess quick IVIM scanning protocols as an additional series to regular clinical acquisition protocols. Quick IVIM protocols may be useful, for example, during the acute stroke phase, where imaging has to be completed in less than 10 min [13], or as a DWI extension for additional screening that provides regular DWI measures, but does not take too much time.

One of the techniques that may help overcome *SNR* issues is voxel clustering or averaging signals from atlas-based regions of interest. Although gray matter exhibits thicknesses of approximately 2–3 mm in the context of brain imaging, in diffusion-weighted images, this tissue effectively occupies nearly the whole voxel between white matter (WM) and cerebrospinal fluid (CSF). The quality of single voxel estimation rather than the clustered approach should also be investigated, as, in this tissue, the precise drawing of large ROIs may be difficult.

To assess the chance of performing an accurate IVIM MRI parameters estimation, the *SNR* level must be known as a crucial factor, determining the reliability and precision of estimation. *SNR* plays a significant role in DWI, influencing the ability to distinguish subtle signal variations related to diffusion from noise in the acquired images. A higher *SNR* allows for better discrimination between actual signal changes related to perfusion effects and random fluctuations. Therefore, knowledge of the *SNR* level is essential for ensuring and estimating the robustness and validity of IVIM MRI analyses.

The primary aim of this study was to investigate the impact of *SNR* on the estimation of the *f*, *D*, and *D** parameters with the IVIM technique. By employing multiple parameter estimation methods, this study aims to compare their efficacy and identify potential strengths and weaknesses in particular *SNR* ranges and region sizes. A simulated environment is created to evaluate the accuracy of the IVIM MRI parameter estimation methods. IVIM parameters were estimated using three methods commonly found in the literature [14], as follows: one-step curve fitting, two-step curve fitting, and the two-step grid search algorithm on simulated ground truth, noise-added data.

## 2. Materials and Methods

Considering the IVIM protocol to be as quick as possible, the scanning duration was limited to 15 min. A MUSE sequence was used to ensure the maximum available *SNR* level [15]. While planning an experiment, we tried selecting *b*-values that were accessible on the scanner and were close to an optimized set [16]. Under these conditions, the GE MR 3T Pioneer scanner with 21 channel head/neck coil was able to acquire images for 10 b-values (1200, 1000, 700, 500, 200, 120, 80, 50, 20, and 10) measured in three orthogonal directions and five $b_{0}$ (no diffusion weighing). Echo Time (TE) was set to 85 ms, and Repetition Time (TR) was set to 14.3 s. Voxel dimensions were 1 × 1 mm in-plane resolution, 2 mm slice thickness, 72 slices (whole brain coverage), slice gap 0, phase acceleration factor was set to 2, and NEX 1 for all *b*-values. Two anatomical sequences were applied, as follows: T1 MP-RAGE TR 1370.36 ms, TE 2.228 ms, TI 718.0 ms, flip angle 7°, 1 mm cubic voxel; and CUBE T2, TE 71 ms, TR 2800 ms, 1mm cubic voxel, reconstructed to 0.5 × 0.5 × 1 mm voxel.

The diffusion data underwent a preprocessing procedure involving the following five steps: denoising, removal of Gibbs ringing artifacts, correction of susceptibility-induced distortions, eddy current correction, and DWI bias field correction. Denoising was performed using the Marchenko–Pastur Principal Component Analysis [17,18,19], while Gibbs ringing artifacts were mitigated through the method of local subvoxel-shifts [20] and were implemented in MRTrix version 3.0.4 [21]; susceptibility-induced distortions correction and eddy current correction was corrected using FSL version 6.0.6.4 [22]; and B1 Inhomogeneity correction was corrected using the N4 algorithm [23] implemented in ANTs version 2.3.1 [24]. Preprocessed data underwent a brain extraction procedure with MRTrix. *SNR* was calculated from 5 $b_{0}$ scans as mean signal and standard deviation quotient in MATLAB version 2023a [25] for brain-extracted images. Normalization of IVIM data to MNI152 space [26] was performed using MRTrix algorithms. To match structures in MNI space, patients’ $b_{0}$ scan underwent extraction, followed by brain masking. Subsequently, the histogram of the extracted image was nonlinearly adjusted to match the histogram of the symmetrical 1 mm^3^ T2 template. The image was then registered to the template using affine, followed by nonlinear transformation. The following two transformation matrices were computed: one for aligning the patient’s image to the template space and another for aligning the template to the patient’s space. Using the abovementioned transform, the Brodmann atlas aligned in MNI space [27] was transformed to patient space and re-gridded to IVIM data using the nearest neighbor interpolation method. The quality of the registration procedure was visually inspected. IVIM parameter estimations for 80 Brodmann areas (40 from the left and 40 from the right hemisphere) were computed based on the averaged signal from the specified region after normalizing each voxel to its maximal value.

FSL FMRIB’s Automated Segmentation Tool was used on T1 and T2 images to create white and gray matter maps. Despite the patient being stable during acquisition, a minimal drift in head position was observed between scans. T1 and IVIM MRI were acquired as the first and the last imaging sequence, respectively, thus the co-registration of T1 to the $b_{0}$ IVIM scan was necessary because of discrepancies in spatial alignment caused by patient movement. This co-registration was performed using SPM12 [28] using the Normalized Mutual Information objective function. Similar to the above, IVIM parameter estimations for WM and grey matter (GM) were computed based on the average signal from the region after normalizing each voxel to its maximal value.

The IVIM signal is generated in silico, mimicking real-world conditions, with known ground truth values assigned to key parameters such as perfusion fraction (f), diffusion coefficient (D), pseudo-diffusion coefficient ($D^{*}$), and baseline signal intensity ($S_{0}$). This allows for the latter comparison of estimated values with the true values, providing a controlled setting to assess the reliability of the estimation methods. These were conducted using MATLAB and included the generation of synthetic IVIM MRI data using the four-parametric model described by Equation (1). According to the literature, in GM *f*, *D*^*^, and *D* are in the range of 2.4–24.7, 6.2–85.7·10^−3^ mm^2^/s, and 0.67–1.20·10^−3^ mm^2^/s, respectively, which was summarized in [29]. To conduct further study, we chose values from the mentioned ranges to represent IVIM parameter values as a sample from a human brain. The following values were assigned as ground truth *S_0_* = 1, *f* = 0.12, *D*^*^ = 0.01 mm^2^/s, and *D* = 0.001 mm^2^/s.

Rician noise is introduced to the simulated IVIM signals to replicate the noise inherent in real-world MRI data. The application of noise ensures that the simulated data resembles the complexities of experimental data, allowing for the evaluation of the parameter estimation quality.

Random Rician-distributed noise, as shown in Equation (2), was added to create a multiple realization of IVIM data. The amplitude of the real and imaginary parts of Rician noise were adequately set to represent *SNR* levels from 15 to 50. Simulations finally consisted of n=17,280 realizations for each *SNR* level.
(2)$$S_{noised}=\sqrt{{\left(S+N_{r}\right)}^{2}+N_{i}^{2}}$$
where:

S represents the “ground truth” signal,$S_{noised}$ represents the noised signal,$N_{r},N_{i}$ are, respectively, real and imaginary parts of noise.

For every realization, parameters were estimated using three methods. The one-step fitting method involves the simultaneous estimation of all parameters, while the two-step methods separate the estimation into two distinct stages. The first stage aims to estimate the regular diffusion coefficient (*D*), for a *b*-value greater than a certain threshold, followed by the second stage, where the pseudo-diffusion coefficient (*D**) is estimated from the remaining data below the threshold. This approach relies on the observation that in data obtained for b-values above 250, the blood fraction signal is close to 0 and below the noise floor level [8]. For both one-step (4-parameter) and two-step (2-parameter) fitting, the Trust Regions nonlinear curve fitting method implemented in the MATLAB Curve Fitting Tool [30] was used. An algorithm was implemented locally for the grid search method.

The grid search algorithm employs maximum likelihood estimation (MLE) for a combination of two parameters. Initially, it only monoexponentially estimates D and $S_{0}$ from the signal obtained for b>250. Subsequently, for b<250, it subtracts the previously estimated signal curve and then fits the remaining data to estimate $D^{*}$ and $S_{o}$ in a similar manner. Dividing $S_{o}$ from the second step by the sum of the $S_{o}$ estimates from both steps produces f. After the estimation procedure, a Root Mean Square Error (*RMSE*) between estimated parameters *Ŝ*_0_, $\hat{f}$, ${\hat{D}}^{*}$, and $\hat{D}$ and known ground truth parameters was calculated for each parameter and *SNR* level. For three analyzed methods, RMSE relative to ground truth was plotted as a function of *SNR*. Additionally, to simulate neighboring voxel averaging, estimates were calculated for every 8 (2 × 2 × 2), 27 (3 × 3 × 3), and 64 (4 × 4 × 4) averaged realizations.

## 3. Results

### 3.1. Estimation of Scanner-Specific SNR

A healthy subject (M, age 55) was scanned using the previously described protocol. The subject gave their informed consent for inclusion before participating in the study. The study was conducted in accordance with the Declaration of Helsinki, and the protocol was approved by the local Ethics Committee for project NCN OPUS 2018/31/B/ST7/01888.

The brain extraction tool found the brain to be occupying 1,011,294 of a total 4,718,592 voxels (approx. 20% of the scanned volume) and *SNR* in the whole brain to have the following parameters: average = 19.57, median = 19.14, minimum = 0.789, and maximum = 150. The results are depicted in Figure 2.

The *SNR* level was separately estimated for WM and GM voxels using FSL FAST segmentation with Tissue Probability Maps (TPMs). Acquiring high-resolution T2 data allowed for the precise separation of these two tissue types. The results were comparable to whole brain analysis and to each other. Estimated parameters for WM, based on 210,523 voxels, were as follows: average = 21.56, median = 21.23, minimum = 1.36, and maximum = 62.14. For GM, mask depicted on Figure 3, based on 68,067 voxels average = 20.69, median = 20.32, minimum = 1.25, and maximum = 117.77.

### 3.2. Evaluation of Accuracy

A comparison of *RMSE* in the function of *SNR* for the three analyzed methods is shown in Figure 4. Further tests included averaging simulation data before estimation for 2 × 2 × 2 (Figure 5), 3 × 3 × 3 (Figure 6), and 4 × 4 × 4 (Figure 7) neighboring voxels, which would be an equivalent of 2 × 2 × 4, 3 × 3 × 6, and 4 × 4 × 8 mm voxels/regions of interest, respectively.

The values of relative *RMSE* for *SNR* 20, closest to the average *SNR* in the conducted study, are presented in Table 1.

### 3.3. Results from Study

#### 3.3.1. Quality of Matching Subject IVIM DWI Data to T2 MNI Template

Fitting to the template was performed using MRTRix. The results are depicted in Figure 8.

#### 3.3.2. Calculation of IVIM Parameters

IVIM parameters were calculated in the following three ways: for WM and GM, for each of the Brodmann areas, and in a single voxel manner. The results for WM and GM are presented in Table 2. The calculation of IVIM parameters for Brodmann areas is summarized using boxplots across all three methods. Three separate plots, containing three boxplots each, were prepared to illustrate the results. Figure 9 presents the blood fraction, pseudo-diffusion, and diffusion estimation results. Each boxplot within a single axes represents estimates from all 80 regions, calculated using the given method. The results of the single voxel calculation are plotted in Figure 10.

Out of 369,832 voxels, estimates from 56,989 (approx. 15.42%) were outliers that reached near 0 blood fraction, f. In total, 70.02% of estimations fell in the range fϵ<0.02; 0.25>. Over 54.82% of the estimates from voxels are considered as outliers. Altogether, 33.21% fell in the range $D^{*}\epsilon <0.006;\;0.05)$.

## 4. Discussion

This study aimed to investigate IVIM estimation procedures in various *SNR* levels in medical conditions. An evaluation of accuracy was performed for commonly used techniques [14]. The SNR level is a factor of great importance whilst predicting IVIM parameters. Unfortunately, it is hard to achieve a high SNR while applying quick scanning techniques. One of the solutions is voxel clusterization. In many IVIM-related papers, this technique is used [10,12,29,31]. Another option for improving *SNR* is using multiple excitations for a single voxel, as mentioned in [16]. In all these works, the sizes of the voxels are much larger than in this paper (i.e., 1.8 × 1.8 × 5 mm [29], 1.2 × 1.2 × 4 mm [10], 2.75 × 2.75 × 5 mm [12], 1.14 × 1.14 × 4 mm [16], and 1.5 × 1.5 × 6 mm [32]) and have a nonzero slice gap or are limited to a lower number of slices, both resulting in worse segmentation capabilities. Although in most publications SNR level is presented to be in the range of 10–30 for $b_{0}$, some publications state SNR is in the range of 35–53 on b-value 1000 [14] or even SNR=50 dB [16], which corresponds with SNR≈316.22, which is very unlikely to achieve in any scanning. To achieve precise results at a high SNR level, one of the works’ total acquisition time was 52 min [12]. Similarly, [32] presents an optimal b vector for a 12 min sequence, but the number of slices is not clearly defined, so the brain coverage remains unknown. This consideration raises the question of how to effectively implement a universal and repeatable IVIM diagnostic sequence.

We showed that high-resolution f and D estimates are consistent with the values reported in the literature for clustered or large voxel studies. Results are more consistent for segmented methods, while the one-step method requires a better *SNR* level. IVIM estimates from a signal averaged over WM and GM ROIs fit well with reports from previous studies. Also, the vast majority of estimations for Brodmann areas fell within the ranges reported in the literature [29].

Some estimated values appear to be outliers, which may stem from small volumes of interest or locally imperfect segmentation. All outliers may also have multiple causes, like imperfect acquisition or patient movement, which was observed between the first and last scan and was properly corrected. Patient movement is another factor emphasizing the importance of keeping acquisition time in tight constraints.

High-resolution IVIM estimates from voxels mainly occupied by vessels, such as small arterials with relatively high blood flow [33,34], may yield high f and $D^{*}$ estimates. Consequently, this could potentially result in categorizing them as outliers.

Our interpretation is that the lower $D^{*}$ estimation than can be seen in most of the literature may result from improved sampling precision in GM and Brodmann areas and, thus, a lower CSF share in the overall signal from the tissue.

Specifically for our scanner, with measured *SNR* = 20, *RMSE* of f and $D^{*}$ on single voxel estimations reached 80–100% and 350–500%, respectively. Achieving a higher *SNR* level near 50 on a better system would ultimately lead to an *RMSE* of a single voxel blood fraction fit near 55%. The *RMSE* of the blood pseudo-diffusion coefficient fit, in all cases, was above 70%, reaching 500–600% for curve fitting methods. These *RMSE* results may suggest that utilizing IVIM in a single voxel manner may not be optimal. On the other hand, combining the signal from 27 1 × 1 × 2 mm voxels in an *SNR* range near that estimated for our scanner for the *RMSE* of blood fraction parameter f estimation was able to achieve the level of 25%, while estimation of $D^{*}$ improved from a 100% relative *RMSE* error to roughly 60%. This improvement of *RMSE* suggests a high potential for good results on ROI-based estimations, which was partially confirmed by estimations conducted on Brodmann areas.

Regarding the selection of fitting techniques, in terms of precise f estimation, it is generally advised to use segmented parameter fitting methods. In all *SNR* cases, grid search and segmented curve fitting represent similar qualities in terms of f prediction, with favor for the grid search algorithm in single voxel fitting. The all-in-one step fitting becomes feasible only for high *SNR* and clustered voxels. In the case of single voxel $D^{*}$ fitting, it is advised to use grid search, as the other methods become reliable either for the clustered voxel approach or at significantly higher *SNR* levels. In low-resolution high *SNR* studies, all three methods will produce results of similar accuracy. In terms of calculation time, employing a one-step method is beneficial, although better precision is typically achieved using segmented methods. The general recommendation is to utilize segmented methods, reserving the one-step method exclusively for studies characterized by high signal-to-noise ratios (*SNR*s).

Fitting may produce imperfect results even for relatively large ROIs, such as Brodmann areas. These outlier values were estimated from the smallest Brodmann ROIs and the frontal cortex, which may be subject to greater movement considering the patient’s supine position.

Acquiring high-resolution IVIM DWI allowed for precise spatial alignment with the template/atlas, as evidenced in Figure 8. Visual observation leads to the conclusion that Brodmann’s atlas is mostly well-aligned with cortical structures.

The decision to average small voxel data rather than acquiring larger voxels may initially appear incongruent with the pursuit of high-resolution IVIM outcomes. However, rather than decreasing spatial resolution, averaging data from multiple precisely selected voxels could improve the diagnostic value of the data. The utilization of smaller voxels proves to be more beneficial, given the fact that with high-resolution imaging sequence, it is possible to draw precise ROIs for any structure. With imaging sequences that have more than a 3 mm slice thickness or in-plane resolution, it is impossible to catch gray matter-specific voxels because of partial volume effects. Acquiring high-resolution IVIM images allows us to distinguish between WM, GM, and CSF and limit the partial volume effect influence on parameter estimation.

This study showed that acquiring high-resolution IVIM data is possible and may also provide precise results if analysis is conducted with care. The presented findings could introduce a new perspective in the approach towards DWI imaging, particularly in the context of clinical scanning. Research may be continued and extended to gather more patient data, search for applications, and test multiple scanners and scanning techniques toward finding an optimal whole-brain quick scanning protocol. Advancements in this field can potentially improve all perfusion-related diagnostics, as well as patients’ comfort and safety.

## 5. Conclusions

The diffusion-based IVIM method undoubtedly offers a non-invasive means to gain insights into blood flow parameters in perfused tissues. In this study, we estimated accuracy using the *RMSE* measure of a quick, high-resolution IVIM imaging protocol with three reconstruction methods under the assumption of not exceeding a 15 min scanning time. We demonstrated that it is possible to acquire high-resolution IVIM data and estimate perfusion parameters from it. Despite the relative *RMSE* of IVIM estimates and results from human studies showing that all single voxel reconstruction methods are significantly error-prone at low *SNR* levels, the measurements still fell within the expected ranges. Although a significant number of outlier values were found, the estimation of f is still feasible in high-resolution quick IVIM, especially when considering the use of small ROIs. In terms of fitting for $D^{*}$, the single voxel method appears to be not applicable. Simulations suggested a high error value, which was confirmed by human study, as over 50% of estimations reach the boundaries of estimation. This study’s findings hold significant implications for optimizing scanning time in MR imaging, potentially paving the way for quicker and more efficient acquisition protocols, which are crucial for practical implementation in clinical reality and also enhance patient comfort and scanner efficiency.

## Figures and Tables

**Figure 1 diagnostics-14-00653-f001:**
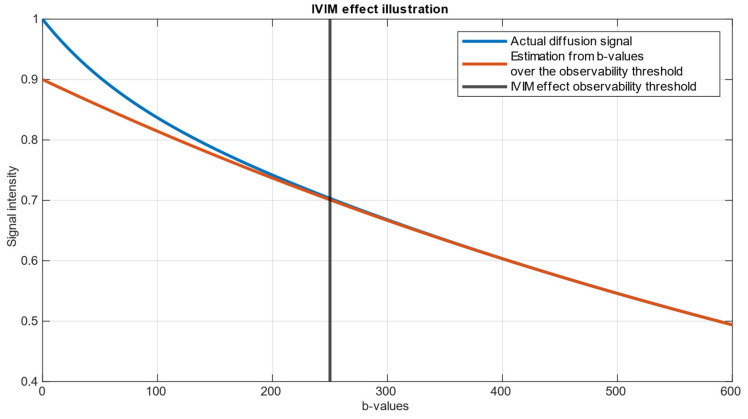
A bi-exponential IVIM model illustration. The orange curve represents the prediction from over-threshold *b*-values. The blue curve represents the actual data. The difference between the prediction from high *b*-values and the actual data is attributed to the influence of blood circulation and is referred to as pseudo-diffusion [8]. It is seen that *b*-values where scanning is sensitive for fast components lie in the range of approx. 0–250.

**Figure 2 diagnostics-14-00653-f002:**
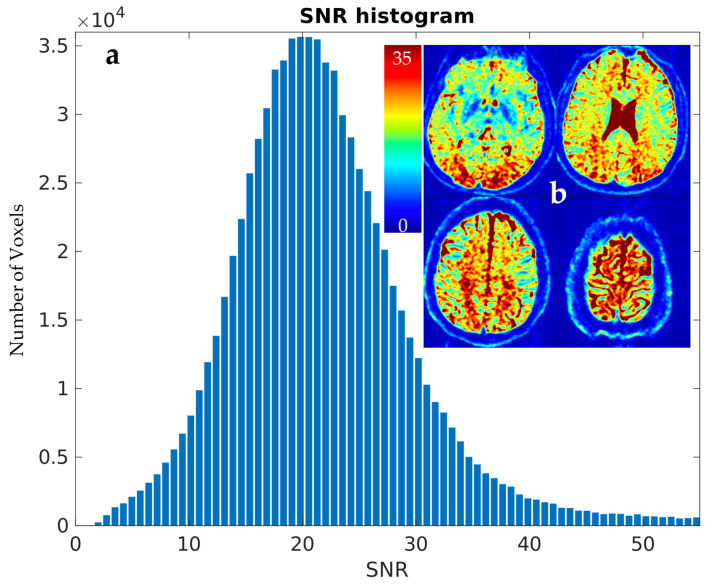
(**a**) Histogram of *SNR* (left). (**b**) Heatmap of *SNR* shown as axial slices 20 mm from each other (right).

**Figure 3 diagnostics-14-00653-f003:**
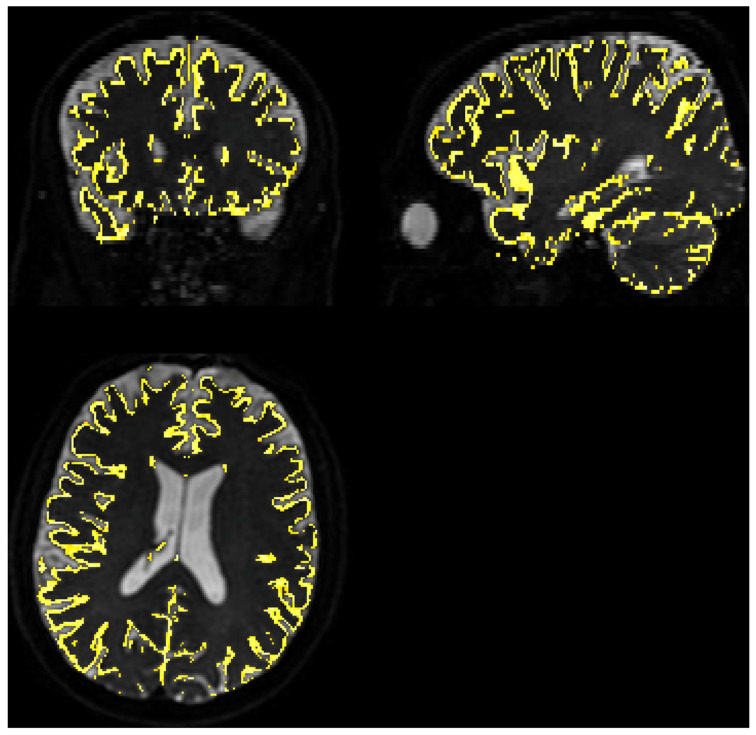
Gray matter coverage on DWI scan. The yellow shape represents the coverage of the gray matter map with a probability > 80% overlayed on $b_{0}$ image, regridded to IVIM DWI resolution.

**Figure 4 diagnostics-14-00653-f004:**
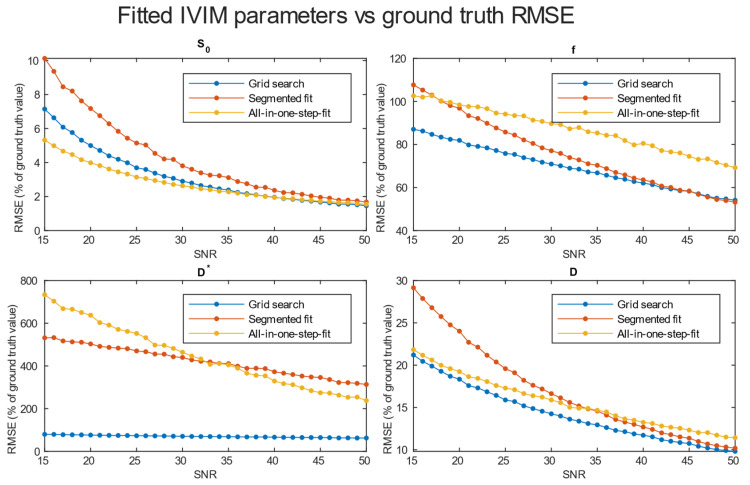
*RMSE* of parameter fitting for three methods of single voxel estimation.

**Figure 5 diagnostics-14-00653-f005:**
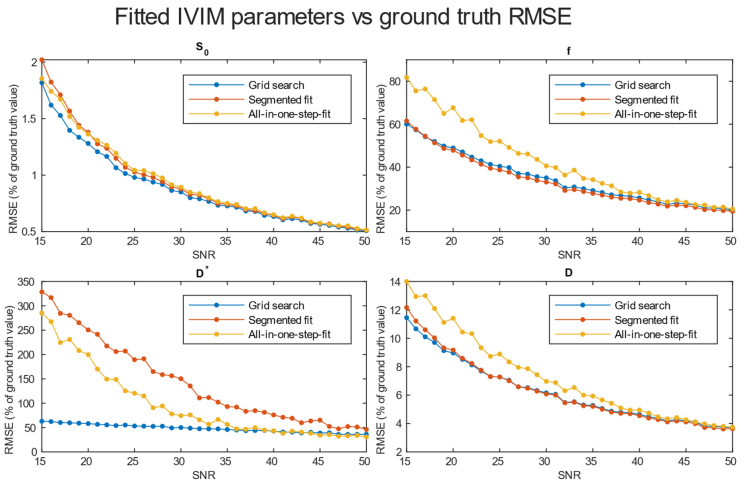
*RMSE* of parameter fitting for three methods. Estimation was made for clustered voxels and each estimation for the signal was averaged from eight voxels.

**Figure 6 diagnostics-14-00653-f006:**
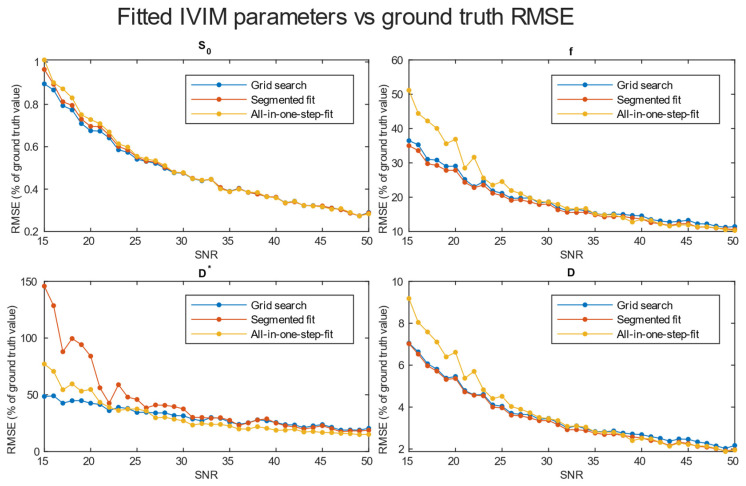
*RMSE* of parameter fitting for three methods. Estimation made for clustered voxels, each estimation for signal averaged from 27 voxels.

**Figure 7 diagnostics-14-00653-f007:**
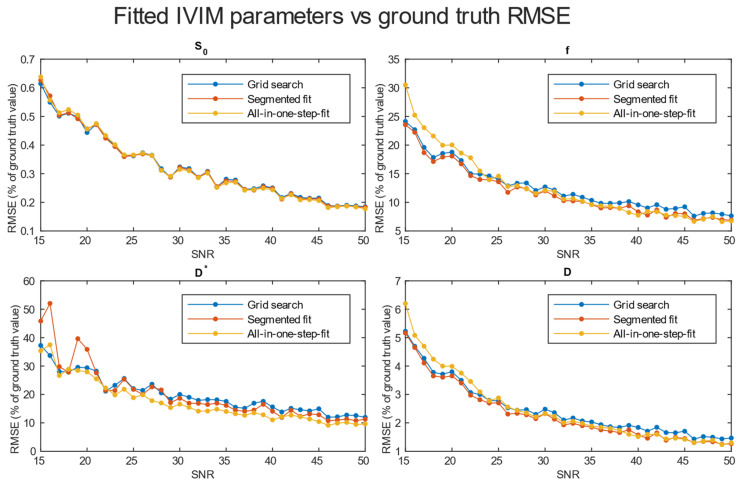
*RMSE* of parameter fitting for three methods. Estimation was made for clustered voxels and each estimation for the signal was averaged from 64 voxels.

**Figure 8 diagnostics-14-00653-f008:**
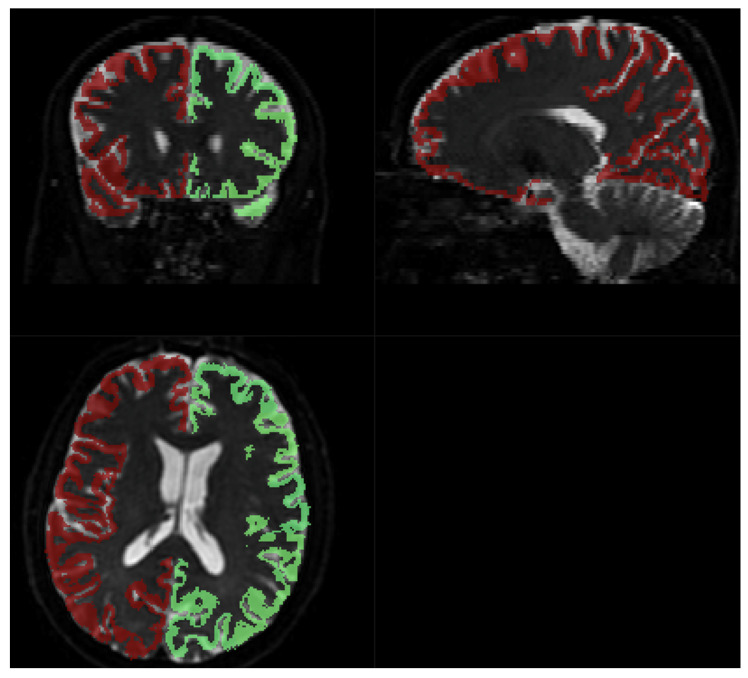
Brodmann areas atlas overlayed on DWI scans. Red and green shape represents the coverage of the Brodmann atlas template, regridded to IVIM DWI resolution.

**Figure 9 diagnostics-14-00653-f009:**
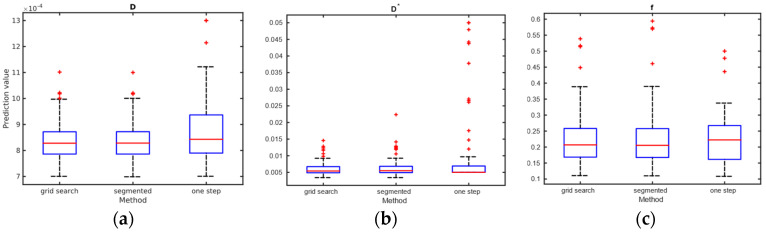
Comparison of (**a**) diffusion, (**b**) pseudo-diffusion parameter estimation in mm^2^/s, and (**c**) blood fraction parameter estimation; calculated on the signal from Brodmann areas.

**Figure 10 diagnostics-14-00653-f010:**
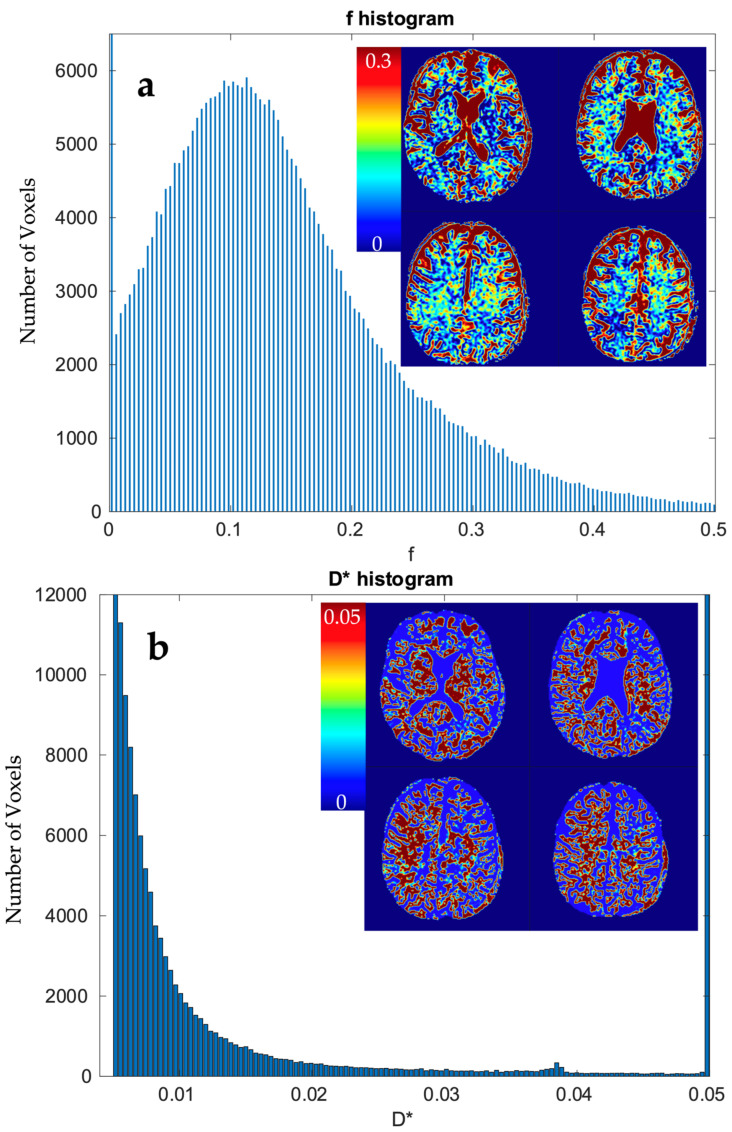
Histogram of estimate value (*x*-axis) on white and grey matter (sum of >70% TPM maps) and heatmap of voxels using grid search calculation method.: (**a**) f and (**b**) $D^{*}$.

**Table 1 diagnostics-14-00653-t001:** Relative *RMSE* error for each estimation method and estimated parameter for *SNR* 20.

Averaging	Grid Search	Segmented	One Step
Single voxel			
*S_0_*	5.00	7.18	3.98
*F*	81.91	96.84	98.40
*D**	76.31	503.04	637.46
*D*	18.34	24.01	19.24
2 × 2 × 2			
*S_0_*	1.28	1.38	1.36
*F*	48.94	47.86	67.67
*D**	58.19	250.51	199.91
*D*	8.97	9.16	11.41
3 × 3 × 3			
*S_0_*	0.68	0.70	0.73
*F*	29.07	27.85	36.89
*D**	42.57	84.06	54.69
*D*	5.46	5.36	6.61
4 × 4 × 4			
*S_0_*	0.44	0.46	0.46
*F*	18.77	18.08	20.02
*D**	29.42	35.88	27.96
*D*	3.80	3.65	3.99

**Table 2 diagnostics-14-00653-t002:** Values of IVIM parameters estimation for white and grey matter for voxels with >80% probability on TPM generated with FSL FAST.

Parameters	Grid Search	Segmented	One Step
White matter			
*f*	0.16	0.16	0.14
$D*\;[10^{-3}\frac{{mm}^{2}}{s}$]	5.17	5.24	5.06
$D\;[10^{-3}\frac{{mm}^{2}}{s}$]	0.79	0.79	0.82
Grey matter			
*f*	0.09	0.08	0.05
$D*\;[10^{-3}\frac{{mm}^{2}}{s}$]	3.62	3.71	5.00
$D\;[10^{-3}\frac{{mm}^{2}}{s}$]	0.65	0.66	0.69

## Data Availability

10.5281/zenodo.10599942.

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
