# Peer review of "Evaluation of Whole Brain Intravoxel Incoherent Motion (IVIM) Imaging"

_diagnostics, 2024, doi:10.3390/diagnostics14060653_

Round 1

Reviewer 1 Report

Comments and Suggestions for Authors

Thank you for performing this study and writing the manuscript. I have some points to remark:

Abstract:

- name the 3 estimation techniques

- sentence in line 17 unclear

Introduction:

- add the word perfusion fraction somewhere in the description of f (lines 51-52)

- Figure 1: Is the blue line the actual data? What is the orange line? The legend is not clear

- reorder the last 3 paragraphs

Materials and Methods

- stay consistent with SNR and signal-to-noise ratio

- move parts to introduction (e.g. lines 91-97, starting line 140, starting line 167)

- add version numbers of used software

- give more information about the grid search method

- remove definition of RMSE, it is known to most people

Results:

- move parts to Methods & Materials (e.g. line 191-194)

- Figure 2: increase resolution of histogram, legend of Figure 2(b) not readable, what is "SNR bin" in y-axis of histogram?

- Figure 3 and 8: remove excessive black background

- Figure 8: chose another color instead of gray

- delete description of box-and-whisker plot. It is known to most people

- make one figure out of Figure 9 and 10

- Figure 11 and 12: axes labels are missing, merge to one figure

Discussion:

- list references for the literature (line 305)

- explain how good or bad the RSME values are (starting line 318)

- highlight the importance of your study

Conclusion:

- rewrite the conclusion. It should be short (one paragraph) and only contain the main findings of your study with a short overview how important your findings are

- consider adding your current Conclusion to Discussion

Comments on the Quality of English Language

Please revise the text grammatically . Pay attention to the use of small and capital letter, commas, articles and word order. The text is sometimes hard to understand. Use the long versions of the abbreviations only in the first use, afterwards not.

Reviewer 2 Report

Comments and Suggestions for Authors

The paper evaluates the feasibility of using the IVIM MRI technique to measure whole brain perfusion parameters in a time-limited clinical setting. A protocol is developed aiming to acquire full brain coverage in under 15 minutes, with 10 b-values and voxels at 1x1x2mm resolution. It concludes that while individual voxel IVIM measures may lack robustness, the overall time-limited acquisition can still provide useful perfusion information when combined with spatial pooling techniques. Trade-offs exist between scan duration, SNR, voxel sizes and need for subject stability. Further optimization of protocols could benefit perfusion diagnostics.

My Major Concerns:

- The study has a small sample size with only one healthy subject scanned. More subjects would allow better generalization of the SNR levels and variability of parameter estimates.

- Motion artifacts caused issues as noted by the authors. Multi-shot sequences may be more robust although slower. The impact of motion is recommended to be investigated or discussed in more detail.

- Selection of the optimal analysis technique is not clear. Each method has different trade-offs that may depend on factors like region size. Recommendations for techniques could be improved.

Comments on the Quality of English Language

The English and presentation of the work are readable but still beneficial for improvement. Such as:

  • Long sentences can be shortened. 
  • Some methods sections are dense with parameters and require re-reading - visual elements like tables may help enhance the presentation.
